



# Single-particle characterization of polycyclic aromatic hydrocarbons in background air in Northern Europe

Johannes Passig[1,2,3], Julian Schade[1,2,5], Robert Irsig[1,2,4], Thomas Kröger-Badge[1,2], Hendryk Czech[1,2,3], Thomas Adam[3,5], Henrik Fallgren[6], Jana Moldanova[6], Martin Sklorz[3], Thorsten Streibel[1,3], and Ralf
Zimmermann[1,2,3]

[1]Joint Mass Spectrometry Centre, Chair of Analytical Chemistry, University Rostock, 18059 Rostock, Germany
[2]Department Life, Light & Matter, University of Rostock, 18059 Rostock, Germany
[3]Joint Mass Spectrometry Centre, Cooperation Group 'Comprehensive Molecular Analytics' (CMA), Helmholtz Zentrum München, 81379 München, Germany
[4]Photonion GmbH, 19061 Schwerin, Germany
[5]Bundeswehr University Munich, 85577 Neubiberg, Germany
[6]IVL Swedish Environmental Research Institute, 411 33 Gothenburg, Sweden

*Correspondence to:* Johannes Passig (johannes.passig@uni-rostock.de)

**Abstract.** We investigated the distribution of polycyclic aromatic hydrocarbons (PAH) on individual ambient aerosol particles at the Swedish west coast in a pristine environment for ten days in October 2019. The measurements were carried out using new technology in single-particle mass spectrometry (SPMS) that reveals both the inorganic particle composition as well as the particle-bound PAHs (Schade et al., 2019). More than 290,000 particles were characterized; 4,412 of them reveal PAH signatures. Most of the PAH-containing particles were internal mixtures of carbonaceous material, secondary nitrate, and
metals from distant sources in Central and Eastern Europe. We characterize the aerosol with respect to the inorganic composition, comparable to conventional SPMS before we discuss the distribution of PAHs within this particle ensemble. Vice versa, we analyze the single-particle PAH spectra for characteristic patterns and discuss the inorganic composition, origin, and atmospheric processing of the respective particles. The study period comprised different meteorological situations: clean air conditions with winds from the North Sea/Kattegat and little terrestrial air pollution, long-range transport from Eastern
Europe and southern Sweden as well as transport of aerosols from Central Europe over the sea. For all meteorological conditions, PAHs were detected in particles whose inorganic content indicates traffic emissions, such as soot, iron, and calcium as well as in particles with biomass burning signatures. However, there were variations in their amounts, dependent on the geographic origin. Because of strong mixing, rapid degradation, and speciation limits, e.g. for PAHs of the same nominal mass, the application of diagnostic ratios for source apportionment is limited under the conditions of our study. Nevertheless, the
combination with the inorganic content and meteorological data provide unique insight into the particles' origin, aging, and mixing state. We exemplarily show how the observation of PAH profiles and inorganic secondary components on a single-particle level can open a new door to investigate aerosol aging processes. To our best knowledge, we herewith present the first comprehensive study on the single-particle distribution of PAHs in ambient air as well as the first set of combined data on PAHs and inorganic composition on a single-particle level.

## 1 Introduction

Polycyclic aromatic hydrocarbons (PAHs) are ubiquitous organic trace components in atmospheric aerosols. They are released into the atmosphere by all types of natural and anthropogenic combustion processes. Because of their well-documented carcinogenicity and mutagenicity, they play a key role in health effects from air pollution (Kim et al., 2013; Agudelo-Castañeda





et al., 2017). After entering the atmosphere, PAHs are widely distributed by aerosols before being deposited onto soils, water,
and vegetation (Ravindra et al., 2008; Dat and Chang, 2017). Estimates of the impact of particular sources, the distribution
pathways, and the degradation processes are crucial for risk assessment. Measurements of PAHs in the atmosphere are typically
based on filter sampling and subsequent analysis via gas chromatography techniques (Pandey et al., 2011; Nozière et al., 2015).
PAHs are emitted as a mixture and their relative concentration ratios are assumed to be characteristic of a particular source
(Ravindra et al., 2008; Tobiszewski and Namieśnik, 2012; Dat and Chang, 2017; Czech et al., 2017). Consequently, pairs of
PAHs with comparable physicochemical properties are often analyzed for source apportionment. However, most of these
diagnostic ratios are not stable, for example, because of different decay ratios for photolysis reactions during atmospheric
aging (Vione et al., 2006; Tobiszewski and Namieśnik, 2012). Also, numerous interactions with other aerosol components that
affect PAH degradation are discussed (Keyte et al., 2013), e.g. shielding of PAHs against oxidants by organic coatings
(Zelenyuk et al., 2017; Shrivastava et al., 2017; Alpert et al., 2021). Detailed information on the PAH-containing particles,
such as their inorganic composition, mixing state or morphology might substantially improve source attribution and
degradation estimates. Beyond that, the occurrence and distribution of PAHs are highly variable as human activity and
meteorological conditions change rapidly. Assessments of the exposure and environmental impact would therefore benefit
from real-time monitoring.

Several aerosol mass spectrometers can obtain chemical information from airborne particles in real-time (Pratt and Prather,
2012; Laskin et al., 2018). Among these techniques, single-particle mass spectrometry (SPMS) stands out for characterizing
individual particles, thus revealing the mixing state of the particle ensemble (Murphy, 2007; Hinz and Spengler, 2007; Riemer
et al., 2019; Passig and Zimmermann, 2020). In conventional SPMS, individual particles are hit by intense laser radiation that
desorbs and ionizes at least a fraction of the particle within a single laser pulse (laser desorption/ionization, LDI). This
ionization technique is highly non-linear; ionization efficiencies of particle compounds vary widely and depend also on the
particle's main components and morphology (Neubauer et al., 1998; Murphy, 2007; Reinard and Johnston, 2008; Hatch et al.,
2011; Hatch et al., 2014). However, LDI produces sufficient ion numbers even from single nanoparticles (Wang et al., 2006)
and is also effective for refractory compounds like metals (Murphy et al., 2006; Dall'Osto et al., 2016b; Arndt et al., 2017;
Dall'Osto et al., 2016a; Passig et al., 2020). Besides its relevance for the environment and health, the refractory components
are typically conserved and bear particle source information that can exclusively be addressed by SPMS.

In contrast to metals or salts, the speciation of organic species in LDI-based SPMS is limited by strong fragmentation of
molecules in the intense UV laser pulse. Only for specific particle matrices such as rather pure carbon (i.e. soot particles),
molecular PAHs are sufficiently detectable by LDI-MS (Zimmermann et al., 2003). Therefore, several approaches for "soft"
ionization of molecules from the particles have been developed, namely single-photon ionization, SPI (Sykes et al., 2002; Nash
et al., 2005; Hanna et al., 2009), and resonance-enhanced multiphoton ionization (REMPI) (Morrical et al., 1998; Bente et al.,
2008, 2009; Li et al., 2019). The latter is especially suitable for SPMS because it is highly specific and very sensitive to PAHs
(Boesl, 2000; Gunzer et al., 2019), thus capable to detect this particularly relevant substance group on a single-particle basis.
The typical REMPI (and SPI) approach is a two-step scheme, where an IR laser pulse heats the particle and produces a gaseous



plume of the organic molecules prior to the UV ionization pulse (laser desorption, LD-REMPI). Vaporization of molecules before ionization also avoids some of the matrix effects associated with single-step LDI and can thus facilitate quantification

(Woods et al., 2001). Only a limited number of such two-step approaches were applied for real-world ambient aerosols (Zelenyuk and Imre, 2005; Bente et al., 2008; Oster et al., 2011) and they have the important disadvantage that the inorganic particle composition from LDI, and thus the particle source information, is lost.

Recently, this limitation was overcome by methods that combine LDI and LD-REMPI (Passig et al., 2017; Schade et al., 2019), detecting positive and negative ions from LDI similar to conventional SPMS together with full-fledged mass spectra of PAHs

via REMPI (Schade et al., 2019). Briefly, a special laser pulse profile is utilized to ionize PAHs after LD from the refractory particle residue, while the latter is hit by a more intense part of the beam, thus inducing LDI and REMPI within a single laser shot. Furthermore, the mass analyzer's transmission is modulated with respect to the different ionization products to achieve sufficient sensitivity and dynamic range.

Here we report on the first field study using this technique and present the first set of single-particle data on PAHs combined

with information on the particle class and origin as derived from the inorganic composition.

## 2 Methods

### 2.1 SPMS instrumentation

The SPMS instrument, its working principle, and parameters have been described in detail previously (Schade et al., 2019) and were not changed. Briefly, the particles are introduced through an aerodynamic lens, detected and sized via light scattering

using a pair of cw-lasers (wavelength $\lambda$= 532 nm) and photomultipliers before entering the mass spectrometer. Shortly before the center of the ion source, the particles are exposed to an IR pulse for laser desorption ($\lambda$= 10.6 µm). Few microseconds later, a parallel UV beam from an excimer laser ($\lambda$= 248 nm) intersects the gaseous plume with moderate intensity ($\sim$3 MW/cm$^2$), ionizing the desorbed PAHs via REMPI. The beam is then back-reflected and focused with a concave mirror and hits the particle residue in the plume center with a high intensity ($\sim$2 GW/cm$^2$) inducing LDI of the refractory components. Cations

from LDI are detected together with the PAH-ions from REMPI in the positive tube of the Time-of-Flight (TOF) mass analyzer and anions from LDI are measured in the negative TOF-tube. To achieve the large dynamic range for measuring the high ion flux from LDI and the much lower signals from REMPI with the same detector, the transmission of the positive flight tube is attenuated by a factor of about 50 for the lighter LDI ions. All ions are extracted with a delay of 0.6 µs after the ionization pulse to improve the quality of mass spectra (Vera et al., 2005; Li et al., 2018). Particle sizing signals and ion TOF data were

recorded by custom LabView software.

### 2.2 Analysis of single-particle mass spectra

A custom software on the Matlab platform (MathWorks Inc.) was used to compute mass spectra from time-of-flight data considering peak area within nominal mass resolution. Mass spectra from positive and negative LDI as well as from REMPI were separately normalized. For clustering, we used the adaptive resonance theory neural network, ART-2a (Carpenter et al.,





1991; Song et al., 1999),  publicly available with the open-source toolkit FATES (Flexible Analysis Toolkit for the Exploration of SPMS data) (Sultana et al., 2017). Of note, the ionization process, as well as the data structure from LDI and REMPI mass spectra differ substantially. PAH patterns appear as peak rows, bearing information in the intensity distribution of many high-mass peaks, which can in most cases not be unequivocally attributed to specific compounds. LDI signals show fewer peaks that are often assignable to particle components. Consequently, we performed independent clustering for LDI and REMPI

mass spectra and characterized the LDI-based clusters concerning their PAH content. Vice versa, we studied the REMPI-based clusters for their inorganic composition via LDI. Clustering parameters for LDI clustering were chosen as 'standard values' with a learning rate of 0.05, a vigilance factor of 0.8, and 20 iterations, while for REMPI clustering, the vigilance factor was 0.7. LDI ion peaks were assigned to the most likely ion at a given mass (m/z).

**2.3 Measurement site and aerosol sampling**

The experiments were performed at a monitoring station in a rural environment at the Swedish west coast, about 35 km south of Gothenburg (coordinates 57°23'37.8"N, 11°54'51.4"E). The site belongs to the Onsala Space Observatorium, which is located on a peninsula with restricted access and nearly no traffic, residential or industrial emissions nearby, see Fig. 1. Ambient air was sampled at 7 m height above ground (20 m above sea level). The low concentration of background aerosols

required enrichment technology. Therefore, an aerosol concentrator, was used (Model 4240, MSP corp., USA) (Romay et al., 2002) that concentrates particles from the 300  l/min intake airflow into a 1  l/min carrier gas stream. After passing a dryer (Model MD-700-12S-1, Perma Pure LLC, U.S.) the aerosols were further concentrated to 0.1 l/min using an additional virtual impactor stage at the SPMS aerodynamic lens inlet (Zhuo et al., 2021). The first concentrator was designed for particles larger than 1 µm, therefore the system's total concentration factor for ambient air particles around 0.5 µm size was estimated to

approximately 10:1 in a previous study (Passig et al., 2020).

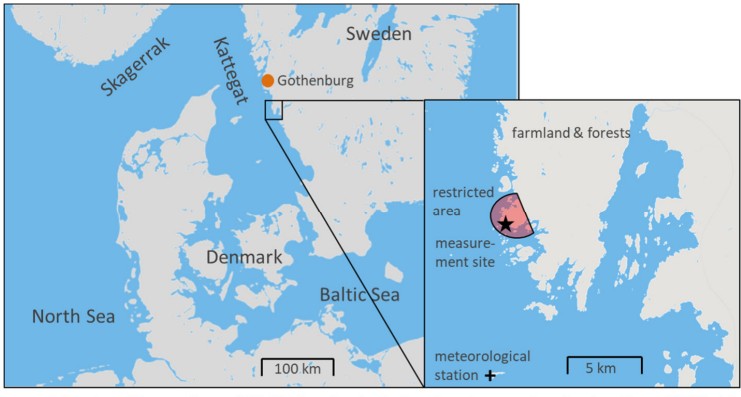

© OpenStreetMap contributors 2020. Distributed under the Open Data Commons Open Database License (ODbL) v1.0.

**Figure 1: Map of the region and position of the measurement site at the Swedish west coast.**



Aerosol mass concentration was measured with a Grimm EDM-180 MC dust monitor that belongs to the standard
instrumentation of the station. Meteorological data was acquired from the Swedish Meteorological and Hydrological Institute,
station Nidingen, located on an island 8 km south of the sampling site (SMHI, 2021). Air back trajectories were computed
using the HYSPLIT web tool from the National Oceanic and Atmospheric Administration, model GFS with 0.25° resolution
(http://www.ready.noaa.gov/HYSPLIT.php, last access 11 January 2021) (Stein et al., 2015).

**3. Results and Discussion**

**3.1 Main particle classes from conventional analysis of LDI mass spectra**

Before focusing on the particle-bound PAHs, we describe the aerosol ensemble according to the usual practice of conventional
LDI-based SPMS. Between the 12th and the 22nd of October 2019, a total number of about 1 Mio. individual particles were
optically detected and sized, whereof 292,242 were analyzed with respect to their chemical composition by at least one mass
spectrum (minimum of four peaks) from either anions from LDI, cations from LDI or PAHs from REMPI. The ART-2a
clustering of LDI signatures yielded 892 clusters. The top 300 clusters account for 86 % of the analyzed particles and were
visually inspected. Clusters with the same overall species and comparable trends and size distributions were manually merged
to 10 general particle classes. Their abundances and mean aerodynamic sizes are shown in Fig. 2(a), while Fig. 2(b) shows
their respective LDI mass spectra. The labelling scheme indicates the most intense peaks and characteristic composition, see
also (Ault et al., 2010; Decesari et al., 2014; Dall'Osto et al., 2016a; Arndt et al., 2017). The mass spectral signatures of the
300 analyzed clusters as well as their temporal behavior are provided in Supplemental Fig. S8 and their assignment to general
classes is documented by Table S1.



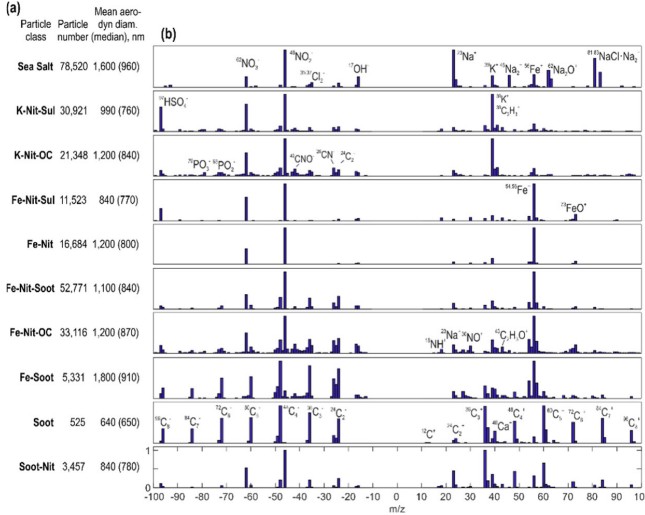


**Figure 2(a) Main particle classes obtained from ART-2a clustering of LDI mass spectra, their particle numbers, and size. (b) Weight matrices (spectra of the cluster center) of negative ions (left) and positive ions (right) corresponding to average mass spectra from LDI. Note that iron is resonantly ionized and therefore efficiently detected. The top 300 ART-2a clusters that were manually merged to these classes are shown in the supplemental Fig. S8.**

The LDI cluster analysis reveals that the aerosol ensemble is dominated by sea salt and by mixtures of carbonaceous particles with nitrate and sulfate. The spectra reveal a relatively high degree of internal mixing (Riemer et al., 2019) compared to SPMS studies in urban regions (Dall'Osto and Harrison, 2006; Healy et al., 2012; Zhang et al., 2013; Giorio et al., 2015; Ma et al., 2016) or marine environments (Middlebrook et al., 1998; Sullivan et al., 2007; Arndt et al., 2017). The strong mixing can be associated with the low contribution of individual local sources, by the geographical location between marine and terrestrial

environments with agriculture, forests, and distant urban areas as well as by high wind speeds that enhances long-range transport. A further reason is the sampling bias towards larger particles with high amounts of secondary material. In the following, the main particle classes are briefly discussed:

Sea salt particles are relatively large and show characteristic signatures from sodium ions (e.g. $^{23}Na^+$, $^{46}NaC^+$, $^{62}Na_2O^+$, $^{63}Na_2OH^+$), $^{39}K^+$, $^{17}OH^-$ and $^{35,37}Cl^-$ (Dall'Osto et al., 2004; Murphy et al., 2019). To keep the number of classes small, we

merged pure sea salt, aged sea salt, and mixtures of sea salt with carbonaceous particles into this class, as long as the salt signatures were dominating and the characteristic signal of $^{81,83}NaCl\cdot Na_2^+$ was clearly recognizable.

Particles with a dominating $K^+$ peak in positive mode are commonly attributed to biomass burning and wood combustion (Silva et al., 1999; Zhang et al., 2013). $CN^-$ and $CNO^-$ peaks in the anion mass spectra were associated with nitrogen-containing organic compounds (Silva and Prather, 2000; Köllner et al., 2017). Here we distinguish between K-Nit particles with strong

sulfate signals from the ones showing larger contributions from organic fragments.





Numerous clusters exhibit strong Fe peaks. Of note, we used a KrF-excimer laser with a wavelength of 248.3 nm, thus the photon energy matches a strong absorption line of atomic iron. As a consequence, iron is detected more universally and with much higher sensitivity than in most other SPMS studies (Passig et al., 2020). Mineral dust was not observed, thus the iron most likely stems from anthropogenic sources such as coal combustion and engine emissions. The iron bound in soot

nanoparticles can enter larger size modes by agglomeration and condensation of secondary material. Most of the clusters with high Fe signals reveal strong peaks of secondary nitrate, indicating distant sources (Furutani et al., 2011; Dall'Osto et al., 2016b). Similar to the K-Nit particles, we found Fe-Nit particles with and without sulfate signature. We also distinguish between particles according to their EC/OC balance (Ferge et al., 2006; Spencer and Prather, 2006), however, this ratio is a continuum in our study (Zhou et al., 2006), manually regrouped concerning the dominant signals.

Larger soot agglomerates were mixed with iron and nitrate, but there was also a limited number of relatively fresh soot particles. They showed $Ca^+$ peaks larger than the $K^+$ (and fragment) signals at m/z=39. It was shown that the majority of soot particles from engines in the accumulation mode reveal $Ca^+$ signatures from lubrication oil additives (Sakurai et al., 2003; Sodeman et al., 2005; Toner et al., 2006). The size distribution of all particle classes is shown in Supplemental Figure S1.

### 3.2 Time series of main particle classes and air mass history

The measurement period was characterized by changing weather conditions with high wind speeds, frequent rain showers, and temperatures varying between 0 °C and 12 °C, during the beginning of the heating season. PM 2.5 data was available between the $14^{th}$ and $21^{tst}$ of October with a mean value of 6.5 µg/cm³, measured with a Grimm EDM-180 MC instrument of the monitoring station. Fig. 3(a) shows the regions the air masses passed within the last 24h before arriving at the sampling site, as obtained from analysis of the HYSPLIT back trajectories shown in Supplemental Fig. S2. Local wind data from the nearby

meteorological station Nidingen is plotted in Fig. 3(b). Particle numbers of the main particle classes with 10 min resolution are shown in Fig. 3(c), while Fig. 3(d) shows the same data normalized to total particle counts.

During western wind directions, marine air is transported from the North Sea via Denmark and the Kattegat, thus the aerosol is dominated by sea salt particles. As apparent from Fig. 3(d) Fe-containing particles with soot are more abundant than particles that belong to the $K^+$-dominated clusters. In addition to remote sources, also marine traffic may contribute. Both the North Sea

and the Baltic Sea are Sulphur Emission Control Areas (SECA), where the large majority of ships run on distillate fuels instead of bunker fuels (Lähteenmäki-Uutela et al., 2019). Consequently, particles with signatures of V and Ni were sparsely detected and are therefore not shown separately from Fe-soot mixtures (Moldanová et al., 2009; Healy et al., 2009; Ault et al., 2010; Passig et al., 2021). Despite the marine origin, only a few sulfate-rich particles were detected, possibly due to the sulfur limits for ship fuels and low marine biogenic activity in autumn.

The highest total particle numbers and smallest contributions from sea salt were observed during eastern winds with air transport from Eastern Europe via the central Baltic Sea and Sweden ($16^{th}$ October). Of note, also the sulfur-containing particles (yellow and cyan) are most abundant during this period, while mixtures with only nitrate dominate outside this period. The sulfur may stem from coal combustion (Xu et al., 2018), which is much more common for residential heating in Eastern Europe





than for Western Europe and Scandinavia. K-Nit-Sul particles are the dominating class, reflecting the importance of residential

wood combustion in Eastern Europe and Sweden.

The 19th and 20th of October were characterized by air transport from Central Europe via the Southern/Western Baltic Sea. Besides sea salt, iron-containing particles with signatures from nitrate, organics, and soot were most abundant, which can be associated with traffic emissions in densely populated European regions (Dall'Osto et al., 2016a) and warmer temperatures in early autumn, where the heating season has not yet begun. Sulfate is now predominantly detected on iron-containing particles,

whereas it was more often associated with potassium-dominated particles during transport from Eastern Europe.

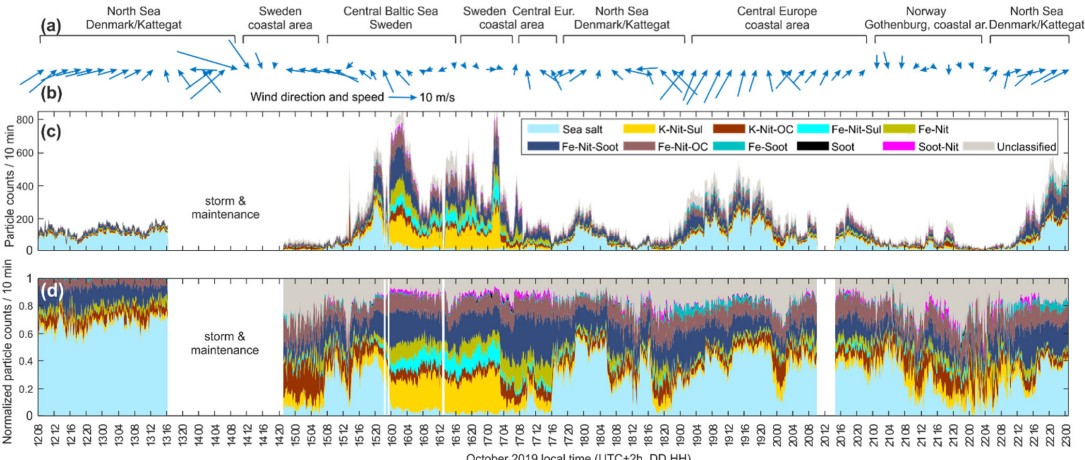

**Figure 3: (a) Air mass origin obtained from analysis of the HYSPLIT back trajectories shown in Fig. S2 (top row: >12 h, bottom row <12 h). (b) Local wind data from a meteorological station 8 km south of the measurement site. (c) Time series of particle counts**

**reveal stable conditions with dominating sea salt during strong wind from the sea and higher particle numbers as well as night-time particle formation events at light winds from land and along the coastline. (d) The same data as (c), normalized to total particle counts reveals the relative contributions of each particle class. High numbers of sulfate-containing, potassium-rich particles can be noticed during air mass transport from Eastern Europe, where the heating season has already started and wood, as well as coal combustion, is common. Wind from Central Europe transports mainly internal mixtures of Fe, organics, and nitrate. Soot signatures are recognized in many particles, but rather pure soot particles and Fe-soot mixtures without secondary material are limited to**

**south-western on-shore winds, pointing to contributions from marine traffic.**

Transient increases in particle numbers can be noticed during low wind speeds on the 16th and 17th of October. Some of them occur at night time, possibly from particle growth into accumulation mode by condensation of secondary material. However, there are also other events such as a green waste burning fire in some kilometers distance in the afternoon of the 15th October and further particle number increases of unknown origin, observed during light wind along the coastline, e.g. on the 17th of

October. Within the first period of marine air and strong wind (12th -14th of October), no transient changes in particle numbers and composition were recognized.



While soot signatures appear in many particles with strong Fe-and nitrate signals during the complete measurement period, the Fe-soot particles without nitrate, as well as fresh soot particles, were only observed during southwestern winds. Assuming that nitrate is mainly secondary from terrestrial emissions, these relative fresh soot particles could be associated with marine

traffic from the major shipping lanes in southwestern direction.

### 3.3 REMPI-PAH signatures in particles classified via LDI

### 3.3.1 General aspects and limitations

Firstly, it should be remembered here, that the peak height in SPMS is not directly convertible to the concentration of a specific substance (Murphy, 2007; Hinz and Spengler, 2007; Reinard and Johnston, 2008; Pratt and Prather, 2012; Healy et al., 2013;

Hatch et al., 2014; Zhou et al., 2016; Gemayel et al., 2017; Shen et al., 2019a). Two-step techniques with separated desorption and ionization, as utilized for the PAHs here, have been shown to be favorable for quantification approaches compared to conventional single-particle LDI (Woods et al., 2001). However, ambiguity remains from different REMPI cross-sections among the PAHs and isomeric structures (Wilkerson et al., 1989; Gehm et al., 2018). Diagnostic ratios for pairs of individual species are typically available for isomeric PAHs (Ravindra et al., 2008; Tobiszewski and Namieśnik, 2012) and are not

directly applicable here. The REMPI cross-sections of degradation products and heterocyclic compounds, such as the health-relevant nitro-PAHs, are typically much lower due to distortion of the $\pi$-electron system (Zimmermann and Hanley, 2020) and can therefore not be detected on a single-particle level.

### 3.3.2 Abundance of PAHs in particles classified by LDI mass spectra

Before analyzing the different PAH patterns found across the particle ensemble, we discuss the abundance and distribution of

PAHs in the main particle classes from LDI. Similar to Fig. 2, the LDI particle classes are again summarized in Fig. 4(a), with an additional column showing the number of PAH-containing particles as determined by the presence of at least four peaks on the m/z channels 178, 189 (fragment of alkylated phenanthrenes), 202, 220, 228 and 252. Fig. 4(b) depicts the corresponding mean mass spectra of PAHs. A first look reveals that sea salt dominated particles show no PAHs, even though this class contains also aged salt particles with secondary nitrate and very low signals of carbonaceous components, compare LDI mean

mass spectra in Fig. 2(b). Harrison et al. found that PAHs from road traffic can be absorbed by salt particles from salting of the road (Harrison et al., 1996). However, for the sea salt particles here, it can be assumed that PAHs were either degraded or in the particulate phase before entering the marine boundary layer and coagulation with sea salt might not be important enough here to detect such mixtures. Also for iron-containing particles with nitrate and/or sulfate, PAH signatures are absent. A general trend becomes already visible here: The stronger the signals of nitrate and sulfate, the lower is the abundance of PAHs in the

particles. There is a number of possible explanations. Firstly, thick coatings of secondary material probably reduce the efficiency of laser desorption for the PAH-containing organic matter (Hatch et al., 2014). On the other hand, it was shown that organic layers can protect PAHs also from photochemical degradation (Zelenyuk et al., 2012; Keyte et al., 2013; Shrivastava



et al., 2017; Alpert et al., 2021), while the effect of secondary nitrate and sulfate on PAH degradation is less investigated. Secondly, high amounts of secondary species suggest a substantial age of the particles, and thus, PAHs may already have been

degraded before. Thirdly, the ionization technique itself may contribute to this trend: Particles that are fully hit by the desorption laser produce more intense PAH spectra via REMPI of the plume. However, secondary nitrate or sulfate is also desorbed to a larger extent from these particles, and less amount of this material remains on the particle for detection via LDI in the second laser shot. Nevertheless, in so-far unpublished field experiments with the same setup during winter in Central Europe, we often found PAHs on more than 20% of all particles, also if strong nitrate- and sulfate signals were present, thus

the instrumental aspect seems to be of minor importance here.

As might be expected, particles with OC fragments in LDI spectra also show PAH signatures in REMPI spectra more frequently, see also the relative numbers of PAH-containing particles within the particle classes in Fig. 4(c). However, the highest PAH abundances can be noticed for soot particles, especially for the relatively fresh and comparably small soot particles. In contrast, the Soot-Nit class shows nearly no PAHs. In addition to some possible effects of the ionization technique,

the PAHs might have been degraded from these aged particles or shielded by thick coatings, as mentioned before. Of note, the classes discussed so far account for only 37% of all particles with PAH signatures. 42% of the PAH-containing particles were not within the top-300 clusters from LDI but have at least 4 LDI peaks, mostly from secondary nitrate, $K^+$, $Fe^+$, and organic fragments, as revealed by a rough screening. About 21% of the PAH-containing particles show less than four LDI peaks. They were not sufficiently hit by the back-reflected LDI laser beam, because numerous organic fragments and $K^+$ would otherwise

appear in the LDI spectrum.

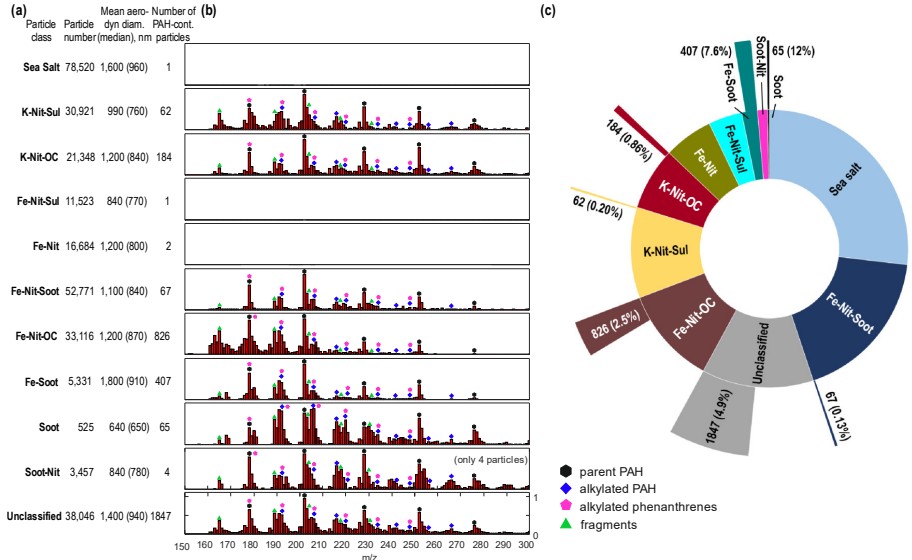

**Figure 4: (a) Main particle classes obtained from ART-2a clustering of LDI mass spectra, their particle numbers, size, and the**
**number of PAH-containing particles in the respective class. Sea salt particles and mixtures with strong secondary nitrate and sulfate**
**show rarely PAHs. (b) Averaged REMPI mass spectra of the respective LDI-clusters. Particles with dominating potassium signatures**
**reveal strong signals of high-mass PAHs, e.g. from wood combustion while particles with prominent Fe-signatures show more**
**alkylated and smaller PAHs, as typical for diesel emissions (Zhang et al., 2008; Tobiszewski and Namieśnik, 2012). Fresh soot**
**particles have higher signals of alkylated phenanthrenes. The unclassified particles were not within the top-300 ART-2a clusters but**
**reveal variable LDI signatures, mostly from organic fragments. Particles with less than four peaks in LDI spectra are not shown**
**here, explaining the missing PAH-containing particles. (c) The inner ring of the sunburst plot illustrates the relative abundance of**
**each LDI-determined particle class. The outer ring depicts the absolute and relative fractions of PAH-containing particles in the**
**respective class. The areas of the outer ring are 10-fold enlarged to make small fractions of PAH-containing particles visible.**

### 3.3.3. Mean PAH signatures in particles classified by LDI mass spectra

When comparing mass spectral patterns of particle-bound PAHs, several aspects beyond the actual experimental details have
to be considered. Previous SPMS studies with LD-REMPI ionization (without information on refractory and inorganic
components) on fresh emissions have identified source-specific signatures in the mass spectra (Morrical et al., 1998; Bente et
al., 2006; Bente et al., 2008, 2009; Li et al., 2019). However, within a multitude of photolysis and oxidation pathways during
atmospheric transport, the different PAHs are degraded with different decay rates (Tobiszewski and Namieśnik, 2012; Keyte
et al., 2013; Pöschl and Shiraiwa, 2015). Therefore, the PAH distributions observed in studies on long-range transported
particles can be substantially different from the patterns observed at a specific source and atmospheric conditions have to be
considered (Tobiszewski and Namieśnik, 2012; Nguyen et al., 2021). The degradation mechanisms are complex and
interfering, however, as a general trend, alkylated PAHs and light-weight species are often reported to be removed more rapidly



(Phousongphouang and Arey, 2002; Lima et al., 2005; Keyte et al., 2013), and differences in the mass-spectral pattern can be
expected to be blurred by degradation processes during long-range transport, diminishing source-specific signatures.

As apparent from Fig. 4(b), unsubstituted 'parent' PAHs contribute the most intense PAH peaks for the LDI-based particle classes, with a maximum for pyrene/fluoranthene (m/z=202), see Table 1. Technical problems were resulting in small inaccuracies of the mass spectrometer and slight broadening of high-mass peaks, however, all key features of the mass spectra remained visible. For the K-Nit-Sul particles and the K-Nit-OC class, the peaks at m/z=228 and m/z=252 from parent high-
molecular-weight PAHs are the second highest and third highest signals. Such distributions are known for pyrogenic PAHs from wood and coal combustion (Czech et al., 2018), in agreement with the source information from LDI. Also gasoline engines emit unsubstituted PAHs of high molecular weight (Miersch et al., 2019), however, particles from engine emissions have a different inorganic composition (Sodeman et al., 2005; Toner et al., 2006; Toner et al., 2008). The PAH spectra of the K-Nit-Sul and the K-Nit-OC class are almost similar. As mentioned before, K-Nit-OC particles reveal PAH signatures more
often, in line with the more pronounced OC fragments in LDI spectra.

Particles with prominent Fe signatures reveal a slightly different mean PAH profile. The PAHs of high molecular weight are less prominent while smaller and alkylated species (e.g. m/z=192 and 206) show a bit more intense peaks, indicative for petrogenic PAHs, e.g. from incomplete combustion in engines (Spencer et al., 2006; Tobiszewski and Namieśnik, 2012; Streibel et al., 2017; Czech et al., 2017). A comparable trend is also documented for PAHs from diesel engines (Spencer et al.,
2006). The increasing fraction of Fe-containing particles during transport from Central Europe (see Fig. 3) is in accordance with the assumption of higher importance of traffic emissions while wood and coal combustion particles were dominant during wind from Eastern Europe.

The relatively fresh soot particles show a different PAH signature with stronger contributions from alkylated phenanthrenes (e.g. m/z=192, 206, 220, 234). With this pattern, they more resemble diesel engine emissions, e.g. from unburned fuel, than
the aged soot particles with nitrate and iron contributions (Spencer et al., 2006; Toner et al., 2006). Most of these particles were detected during on-shore wind, hence ship traffic is the most likely source. The fewer particles observed at eastern wind may stem from local wood combustion and land-based traffic. The PAH signatures of these sources are mixed in the depicted mean spectrum.

A substantial fraction of PAH-containing particles is not classified concerning their inorganic composition because they belong
to clusters with small particle numbers and high orders beyond 300, see the bottom row in Fig. 4(b). Their mean PAH spectrum is mixed from different particle types and possible differences between PAH spectra are averaged, emphasizing the need for a single-particle analysis of the PAH spectra.






**Table 1: Polycyclic aromatic hydrocarbons, indicated by the REMPI mass spectra (m/z).**

| PAHs | number of C in aliphatic side chain(s) | | | | |
|---|---|---|---|---|---|
| | 0 | 1 | 2 | 3 | 4 |
| naphthalene | 128 | 142 | 156 | 170 | 184 |
| acenaphthene | 152 | | | | |
| phenanthrene, anthracene* | 178 | 192 | 206 | 220 | 234 |
| pyrene, fluoranthene | 202 | 216** | 230*** | 244**** | |
| Chrysene, benzoanthracene, benzophenanthrene(s), triphenylene | 228 | 242 | | | |
| benzopyrene(s), benzofluoranthene(s), perylene | 252 | 266 | | | |
| benzo[g,h,i]perylene, indeno(1,2,3)[c,d]pyrene | 276 | | | | |
| dibenzophenanthrene(s), dibenzoanthracene(s), benzochrysene, picene | 278 | | | | |
| important fragments | 115, 139, 165, 189 | | | | |

\*      alkylated species predominantly belong to the homologuous series of alkylated phenanthrenes

\*\*      possible interference with benzofluorenes and cyclopentaphenanthrenes

\*\*\*      possible interference with terphenyl

\*\*\*\*      possible interference with benzylbiphenyl

### 3.4 Single-particle analysis of PAHs

In the previous analyses, the individual PAH spectra were merged within LDI particle classes that represent their inorganic
composition, yielding rather subtle differences among the mean PAH spectra in Fig. 4(b).

To unravel the single-particle PAH composition of the ensemble, we performed a further ART-2a analysis exclusively for the
PAH spectra from REMPI. The analysis with a vigilance factor of 0.7 yielded 733 clusters, whereof the top 300 were manually
inspected, see Supplemental Fig. S9 for all 300 cluster spectra. High-mass inorganic compounds such as lead or potassium
phosphates also appear in the respective m/z range but can be easily excluded from the PAH analysis by their clearly different
signatures. The top 300 clusters include 3746 (84.9%) of in total 4412 PAH-containing particles. The 300 clusters were then
manually merged to 10 PAH classes with respect to their main mass spectral characteristics, see Table 2.






**Table 2: Main particle classes obtained from ART-2a clustering of PAH mass spectra, their particle numbers, and size. HMW: high molecular weight (four rings or more), LMW: low molecular weight (less than four rings), parent: unsubstituted parent PAHs, OC: organic aside from PAHs. For all cluster spectra see Supplemental Fig. S9 and Supplemental Table S2.**

| PAH class | 'common' | 'PAH-HMW' | 'parent' | 'parent-HMW' | 'PAH-OC-LMW' | 'PAH-OC-HMW' | 'OC-HMW' | 'alkylated' | 'alkylated-LMW' | 'fragmented' |
|---|---|---|---|---|---|---|---|---|---|---|
| Particle number | 1413 | 161 | 251 | 225 | 650 | 87 | 176 | 168 | 71 | 544 |
| Median aerodynamic diameter (nm) | 669 | 558 | 644 | 573 | 872 | 656 | 942 | 369 | 525 | 630 |
| Figure | 5 | 6 | S3 | S4 | 7 | 8 | S5 | 9 | S6 | S7 |
| Context source/process | Highest abundance, strong mixing | Biomass burning | PAH degradation | PAH degradation | Secondary organic | Aged POA/ Oligomeri-zation | Aged POA/ Oligomeri-zation | Ship traffic | Localreen waste burning | unknown |


In the following, we discuss the most important PAH particle classes and approach the full depth of single-particle information to identify signatures of sources and atmospheric processing in the particle ensemble.

Fig. 5 (a) shows the mass spectrum of the most abundant PAH profile (n=1,413 particles), hereinafter referred to as the 'common' cluster. It is dominated by unsubstituted ('parent') PAHs and peaks for m/z=202. Also, the aforementioned fragment

at m/z=189 and smaller signatures of alkylated PAHs (e.g. m/z=206) can be noticed. The averaged LDI spectra from the respective particles (Fig. 5(b)) show strong contributions from EC, $K^+$, $Ca^+$, $Fe^+$, smaller OC peaks, and minor nitrate signals. In Fig. 5(c), all 3,746 PAH-classified particles are shown as grey dots, positioned with respect to their detection time (x-axis) and vacuum aerodynamic diameter (y-axis). The red dots represent the particles of the actual cluster and reveal that particles of the 'common' type appeared during all periods of the measurement, however with increased abundances on November

15th/16th during air transport from Eastern Europe, compare Fig. 3.





**Figure 5: (a)** Weight matrix (spectrum of the cluster center) of the most abundant 'common' PAH class (1,413 particles, whereof 1,119 show additional LDI mass spectra with at least four peaks, med size: median vacuum aerodynamic diameter). **(b)** Averaged LDI mass spectra of the particles that belong to this cluster. **(c)** Occurrence of these particles (red dots, n=1,413) within the measurement period (x-axis) and their vacuum aerodynamic diameter (y-axis). Grey dots: PAH-containing particles from the other PAH classes 2–10 (n=2,333). **(d)** Left panel: number of particles in the subgroups as a result of sub-clustering the LDI signatures of particles in the 'common' PAH class. Here, the EC-OC particle type dominates. Right panel: averaged relative peak areas of the key inorganic components for the particles in the subgroups. (Ion mass channels: EC: 24–84/36, 48; OC: 42/27, 29, 55, 63; Ox: 45, 59, 71/43; K: -/39; Nit: 46, 62/-; Sul: 97, 99/-; Fe: -/54, 56; Ca: -/40; Phos: 63, 79/- for negative/positive m/z, respectively).

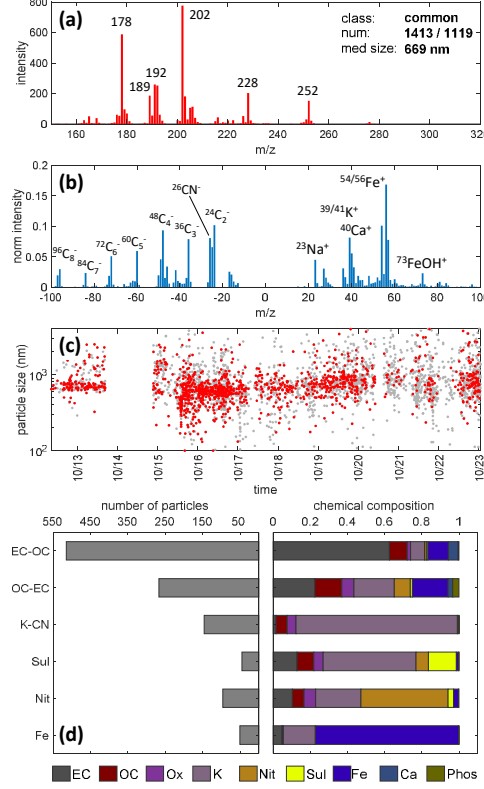

To untangle the interrelations between PAHs, the particles' inorganic composition, and atmospheric processing, we performed a sub-analysis of the particles' inorganic composition for the 10 PAH types. This sub-clustering of LDI signals from the particles of each PAH-class was performed with a vigilance factor of 0.7 and the results were manually merged to six subgroups, respectively. The result for the 'common' PAH-particle type is depicted in Fig. 5(d). The left panel shows the number of particles in each subgroup and the right panel the averaged inorganic composition of the particles in the respective subgroup. For example, this allows comparing the chemical composition of the EC-OC subgroup of particles belonging to one of the PAH classes to the respective EC-OC subgroup of another one of the 10 PAH classes.

It is impossible to discuss all details in this unprecedented dataset, so we focus on our most important findings and refer to the Supplemental Figs. S3–S7, where corresponding plots for further PAH classes can be found.

The majority of particles with a 'common' PAH profile show dominant EC signatures (Fig. 5(d)), while also all other particle subgroups occur. Both $Fe^+$ (traffic, industrial) and $K^+$ (if dominant: biomass burning) signatures can be recognized among the most particle subgroups, indicating that this PAH pattern is not indicative for some specific source but rather a result from extended mixing and atmospheric processing. This is in line with the typical size of the particles (Fig. 5(c)) well above common sizes of fresh soot emissions and with the ubiquitous occurrence during the entire measurement period; both speak against local sources.





### 3.4.2 Traffic and biomass burning source profiles

The PAH cluster shown in Fig. 6 is characterized by a peak pattern that extends to much higher masses compared to the 'common' class in Fig. 5. Such particles were occasionally detected during the full measurement period, with increased numbers in the evening of November 19th and 21st. The mean LDI spectra (Fig. 6(b)) and the single-particle inorganic composition (Fig. 6(d)) reveal that EC-particles are the most abundant group showing this PAH profile and OC fragment signals are rather small. In contrast to the 'common' PAH class in Fig. 5, the positive mass spectra are dominated by the $K^+$

peak, while $Fe^+$ is virtually absent, pointing on wood/biomass burning sources (Lee et al., 2016; Dall'Osto et al., 2016a).

High ratios of HMW/LMW-PAHs have been discussed as an indicator for both fossil fuels and wood combustion (Zhang et al., 2008; Ravindra et al., 2008; Tobiszewski and Namieśnik, 2012). For the long-range transported particles in our study, strong signals

of HMW PAHs are consistently associated with wood/biomass signatures from LDI. Both the volatilization of LMW PAHs and their rapid degradation in the gas phase (Lima et al., 2005; Keyte et al., 2013) as well as possible shielding effects from internal mixing with low-volatility biomass burning organic aerosol (Shrivastava et

al., 2017; Alpert et al., 2021) can be assumed to play an important role here.

In conventional LDI-based source apportionment in SPMS, biomass burning particles are typically identified by their $K^+$ and

$CN^-$ peaks (Dall'Osto et al., 2016a). An important finding here is that additional PAH information can help to identify the EC particles from biomass combustion. Coal combustion may contribute a comparable PAH pattern (Xu et al., 2018) but because of the absence of peaks from Fe and other transition metals, it is an

unlikely source here.

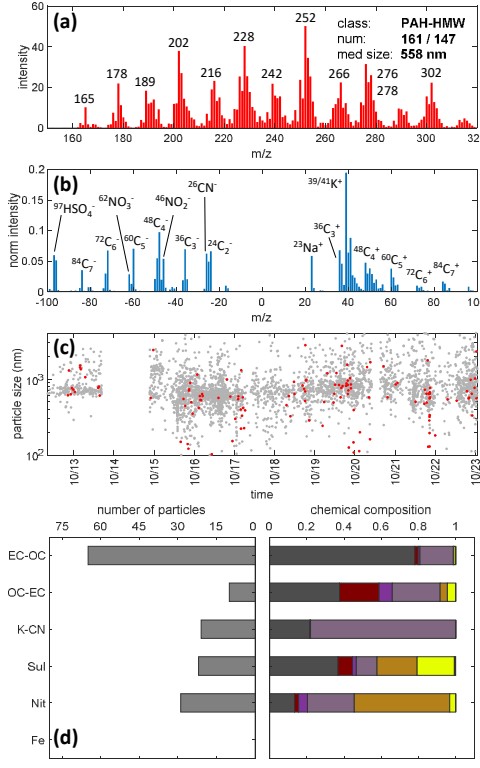

**Figure 6: PAH patterns up to high molecular weights are associated with strong $K^+$ signals and the absence of $Fe^+$, giving evidence for biomass burning particles. (For figure explanation, see Fig. 5.)**



### 3.4.3 PAH degradation

Two PAH clusters reveal only the parent PAHs, while further peaks from fragments and substitutes are absent, see Supplemental Figs. S3 and S4. The inorganic composition of particles from the 'parent' PAH cluster shown in Fig. S3 resembles the 'common' cluster (Fig. 5), however with increased contributions from nitrate. The nitrate signatures of secondary material are further enhanced for the 'parent' PAH cluster with high molecular weight PAHs (Fig. S4). The shift towards high-mass PAHs can again be associated with the biomass burning contribution by the absence of Fe and the dominance of K in the particle's inorganic composition. However, in direct comparison to the 'HMW' PAH cluster in Fig. 6, the particles in the 'parent HMW' PAH cluster in Fig. S4 show more frequently strong signals of secondary nitrate and sulfate. Thus, the observation of peaks from solely unsubstituted PAHs may indicate PAH degradation, as the alkylated PAHs tend to have shorter half-lives (Garrett et al., 1998; Lima et al., 2005) and some of the larger PAHs are more stable against photo-oxidation processes in the particulate phase. (Keyte et al., 2013). Although PAH degradation in airborne particles is strongly affected by matrix effects and complex heterogeneous chemistry, such evaluation of PAH signatures may provide a rough estimate on degradation, if the source profile is known or indicated by the inorganic particle composition in a sophisticated future measurement scenario.





### 3.4.4 Secondary organic material

Several PAH classes are characterized by rows of peaks in addition to the PAH signatures. The PAH spectrum in Fig. 7(a) resembles the 'common' type (Fig. 5(a)), however, additional peaks in the low molecular weight range are clearly visible. The average LDI spectra in Fig. 7(b) shows slightly higher fragment signals compared to Fig. 5(b); in particular, the peak at m/z=43 – an established marker for oxygen-containing organic material due to the contribution of $^{43}C_2H_3O^+$ – is increased (Silva and Prather, 2000; Köllner et al., 2017). The particles of this PAH cluster were larger on average and they were more frequently detected in air masses with a terrestrial background compared to marine air (Fig. 7(c)). In conjunction with the high number of nitrate-containing particles in this PAH-cluster (Fig. 7(d)), we assume a high amount of secondary organic material and secondary nitrate for this type of particles (Shen et al., 2019b; Huang et al., 2019).

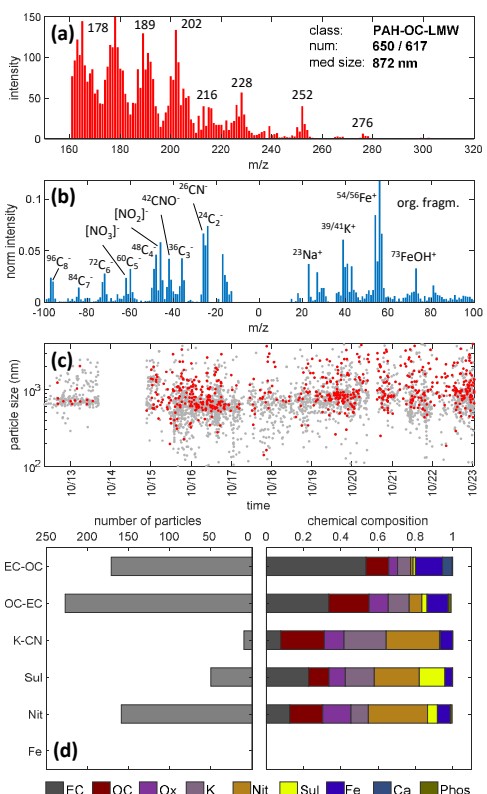

**Figure 7: Additional peaks from OC, relatively large particle size, and frequently detected $^{46}NO_2^-$ and $^{62}NO_3^-$ signals indicate the uptake of secondary material. (For figure explanation, see Fig. 5.)**

### 3.4.5 High molecular weight organics and oligomer formation

The PAH-cluster shown in Fig. 8 reveals PAH peaks and organics up to higher molecular weights (PAH-OC-HMW). The inorganic composition differs from the PAH-OC-LMW cluster (Fig. 7) in two major aspects: Firstly, Fe-signatures are smaller and can only be found in some EC-dominated particles. This can explain the HMW-PAHs, which were already associated with biomass burning sources under the conditions of our study (Sec. 3.4.2). Secondly, the particle's secondary composition reveals strong sulfate signals in addition to the nitrate. The HMW-PAH cluster in Fig. 6 shows a comparable distribution of PAHs but fewer other organic signatures and less sulfate, suggesting a link between sulfate and HMW-organics. Accelerated oligomer formation by particle acidity is well established (Gross et al., 2006; Denkenberger et al., 2007; Wang et al., 2010; Riva et al., 2019), however, also the presence of PAHs itself was found to affect oligomer formation (Zelenyuk et al., 2017; Vereecken,





2018). The small particle number in our experiment allows no statistically sound analysis of the interrelations between these aerosol components, however, more systematic and sophisticated studies based on our approach will provide deeper insights. Of note, oligomerization refers not to the PAHs themselves in our study, as their concentration is probably too low and the resulting signatures would be expected in the mass range beyond m/z=300. Consequently, we assume that non-aromatic oligomers and heterogenic reactions between the PAHs and e.g. OH- contribute to these signatures ('aged primary organic aerosol, POA').

A further cluster with HMW organics and weaker PAH signatures is shown in Fig. S5. The inorganic composition of particles in this OC-HMW cluster shares the high nitrate and sulfate signals of the PAH-OC-HMW cluster in Fig. 8, however, the EC signatures are nearly absent, emphasizing the close connection of PAHs to EC.

**Figure 8: High molecular weight organics in addition to PAH peaks are frequently associated with fragments from oxidized OC (e.g. $^{43}C_2H_3O^+$) and high sulfate signals. This supports the promoted SOA formation and oligomerization in acidic particles. Note also the cluster with dominating HMW-OC and high sulfate signals in Fig. S5. (For figure explanation, see Fig. 5.)**

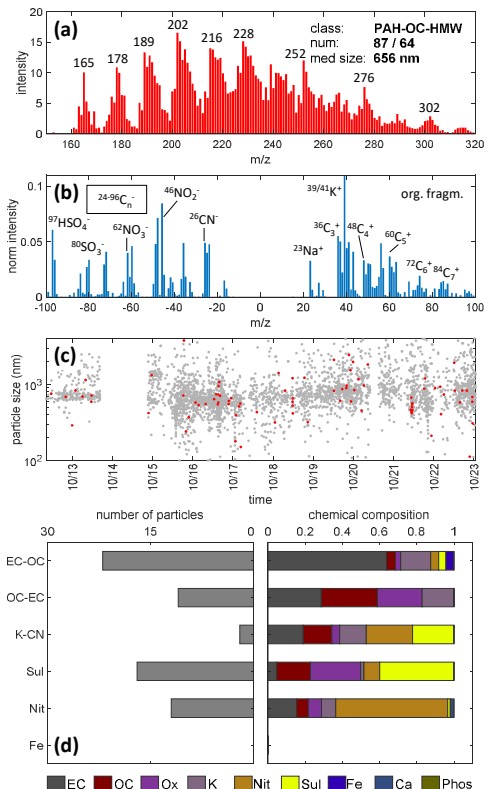

### 3.4.6 Local sources

In addition to the aged and mixed particles from remote regions, the cluster analysis of PAH spectra revealed also signatures that can be attributed to local sources. The aforementioned green waste burning fire on the 15th of October (see Fig. 3) produced a particle type with solely LMW-PAHs and a high peak of a methylphenanthrene at m/z=192, see Fig. S6.

A unique signature was occasionally observed during onshore wind, with a PAH spectrum that is dominated by alkylated phenanthrenes, see Fig. 9(a). The inorganic composition of these comparable small particles is characterized by strong EC signatures while signals from nitrate and sulfate are rarely detected, indicating rather fresh emissions from a nearby source (Fig. 9b-d). Furthermore, this is the only particle type with strong Ca$^+$ signals. The combination of EC and Ca indicates combustion engine emissions, which frequently exhibit Ca$^+$ signals from lubrication oil additives (Toner et al., 2006). These particles can be associated with marine traffic at the main shipping lane in approx. 30 km distance. The PAH profile with dominant signals from alkylated phenanthrenes that peak around m/z=206 was previously in the analysis of volatile

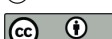



to semi-volatile aromatic compounds in ship emissions (Czech et al., 2017). Of note, the entire Baltic Sea is a sulfur emission control area, where conventional high-sulfur bunker fuels are prohibited and most ships are operated with distillate fuels (Lähteenmäki-Uutela et al., 2019). Thus, the characteristic metal and sulfate signals from residual fuels are not observed here (Healy et al., 2009; Ault et al., 2010; Passig et al., 2021).


A further cluster exhibits a characteristic fragmentation pattern with additional peaks at -2 m/z for each PAH signal, see Fig. S7. While this fragmentation pathway is documented for PAHs (Kruth et al., 2017), the correlations of these signatures with the inorganic composition or atmospheric conditions could not be explored so far and will be subject to a future investigation.

**Fig. 9: Particles with dominant signals from alkylated phenanthrenes were observed during onshore wind periods. These rather small particles reveal soot and Ca⁺, but nearly no secondary material. They can be attributed to ship traffic with distillate fuels such as marine gas oil. (For figure explanation, see Fig. 5.)**


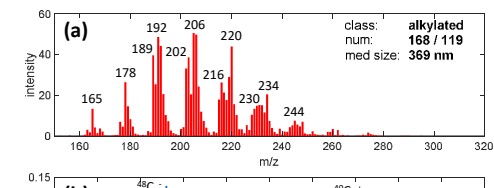


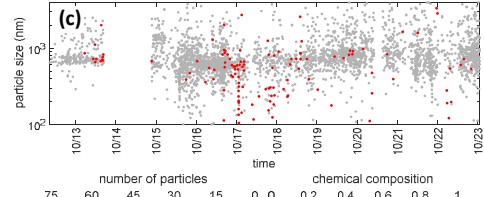

## 4. Conclusions

In summary, we described the single-particle mass-spectral profiles and distribution of PAHs in ambient air in a rural coastal area in autumn. Therefore, we presented two different approaches to tackle this complex and multi-dimensional dataset. Firstly, we classified the particles concerning their inorganic composition and identified the major carriers of PAHs. We found that the averaged PAH profiles are relatively uniform for most of the particle classes and attributed this to mixing and PAH degradation during atmospheric transport. However, the source-specific inorganic composition of the particles could be linked with pyrogenic and petrogenic PAH signatures. Secondly, we classified the particles with respect to characteristic PAH patterns and provided single-particle information on the inorganic composition of the respective PAH




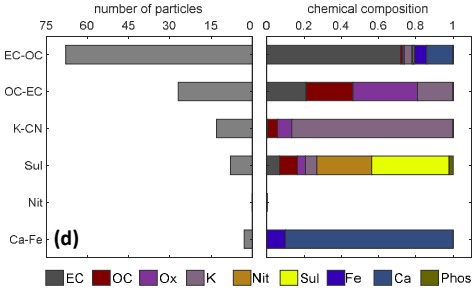

classes. Thereby, a number of distinct PAH profiles could be identified. We associated them with different particle sources and important atmospheric processes, including PAH degradation, secondary aerosol formation, and oligomerization. Here, a critical step has endeavored as some of these interpretations are provisional because of the limited number of particles, the lack of reference data, and the novel type of data without precedent. Therefore, we desist from extensive correlation analyses



and statistics, provide as much insight into the chemical data as possible and try to leave room for interpretation by the reader and an open discussion. Future studies including quantification approaches and simultaneous operation of established on-line technologies such as the Aerodyne AMS (Canagaratna et al., 2007) are planned.

The presented data provide a snapshot of the PAH pollution in a specific region and timeframe, where the conditions were characterized by relatively clean air and dominant long-range transport. However, the study is the first of a series of such

measurements that will shed a light on the respective situations in different regions and seasons. While the PAH burden appeared relatively low here, with only about 1.5% of the particles showing clear PAH signatures, we found PAHs on more than 20% of analyzed particles during extended periods of a winter measurement campaign that will be published in a follow-up paper.

### Data availability

Data are available on request from Johannes Passig (johannes.passig@uni-rostock.de).

### Author contributions

JP and JS contributed equally to this work. JP conceived the study. JS performed the experiments with support from RI, TKB, TA, MS, TS and HF. JS and JP analyzed the data and prepared the figures. HC and RZ provided assistance with data interpretation. JM and HF hosted and supported the field study. JP wrote the manuscript with contributions from all authors.

### Competing interests


The authors declare that they have no conflict of interest.

### Acknowledgements

We thank Johan Mellqvist, John Conway, Lars Eriksson and co-workers from the Chalmers University of Technology and from the IVL Swedish Environmental Research Institute for hosting the field experiments and their support.

The authors gratefully acknowledge the NOAA Air Resources Laboratory (ARL) for the provision of the HYSPLIT transport and dispersion model and READY website (https://www.ready.noaa.gov, last access: 20$^{th}$ August 2021) used in this publication.



**Financial support**

This research has been supported by the Helmholtz-Gemeinschaft (International Lab aerohealth (Interlabs-0005) and Virtual
Institute of Complex Molecular Systems in Environmental Health, HICE), by the Deutsche Forschungsgemeinschaft (grant
no. ZI 764/6-1) and the Bundesministerium für Wirtschaft und Energie (grant no. ZF4402101ZG7 and 16KN083626).

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
