# Peer review of "Single-particle characterization of polycyclic aromatic hydrocarbons in background air in Northern Europe"

_Atmospheric Chemistry and Physics, 2021_

## Author Comment (AC1)

Response to Anonymous Reviewer #1
We thank the referee for reviewing our manuscript and the valuable comments. Please see our reply below.

Note:
*Reviewer comments are in italics.*
Author responses are in normal format.
**Changes** that were made to the manuscript are in **bold** face.
* * *
*Review of "Single-particle characterization of polycyclic aromatic hydrocarbons in background air in Northern Europe"*

*Passig and coauthors describe a measurement campaign at a remote site in coastal Sweden using a two-step LDI/REMPI single particle mass spectrometer designed to measure inorganic, general organic and more specific PAH particle content. Because the instrument separately records more typical LDI mass spectra and more unique REMPI PAH mass spectra, two approaches were taken to bring this compositional information together. Firstly, the LDI mass spectra were clustered using the regular neural network approach, and the corresponding REMPI PAH signatures were examined to gain insight on sources/processing. Secondly, the REMPI dataset was clustered and within those resulting PAH classes the LDI mass spectra were further examined to highlight differences between sources and processes for PAH-containing particles. Although PAHs were detected in relatively few particles in the overall dataset (which can be explained by the remote location and the dominance of aged particles), some interesting connections between PAH mass spectral patterns and the refractory core composition of the single particles can nonetheless be obtained. While the article represents a 'proof-of-concept' for the method, the features of the dataset are useful for informing future particle classifications at other sites. A few things stand out. Firstly the connection between potassium-rich particle cores and pyrogenic PAH signatures (m/z 228/252) is emerging as a useful signature for woodburning particles, even after long atmospheric processing times. Secondly, the connection between iron (detected with high sensitivity using the excimer laser here) and petrogenic PAH signatures (m/z192/206) is useful for identifying aged engine exhaust. Thirdly, more fresh soot particles associated with engine exhaust are characterized by LDI signals for calcium and alkylated phenanthrenes. This information is useful for identifying the original primary sources of these particles, even at remote sites. The discussion of limitations is also useful, as there remain potential hurdles for this type of analysis, including the difficulty in measuring substituted or heterocyclic PAHs. Overall, however, as a first examination of how to parse the complex ambient datasets generated using this new approach, the findings here are valuable for the single particle and source apportionment communities. I only have minor comments to suggest.*

We thank the referee for the comments and the vital discussion.

*Line 90 define continuous wave*

Thank you. **Corrected.**

*Line 91: "A few"*

**Corrected.**

*Line 114: It would be helpful to discuss how the vigilance factor was arrived at for the REMPI dataset*

The vigilance factor of 0.7 was determined in tests of the clustering and manual checks. We added a comment on that: **"…a value that resulted in a good balance between the number of clusters that can be evaluated in manual regrouping and the recognition of prominent PAH signatures.**

*Line 139: M for million*

**Corrected.**

*Line 163: $Na_2^+$*

We apologize for the error. **Corrected accordingly.**

*Line 166: $Na_2Cl$ for m/z 81/83*

We apologize for the error. **Corrected accordingly.**

*Line 179: Was this EC/OC regrouping used to inform the final 10 classes in some way?*

The final 10 classes were realized by manual regrouping after ART2a, as stated earlier in the manuscript. There was no additional regrouping for the EC/OC ratio. **We removed the misleading statement from the manuscript.**

*Line 186 subscript 2.5*

**Changed accordingly.**

*Line 187 $m^3$*

Thank you for careful check! **Corrected.**

*Figure 3 part of caption unclear: top row and bottom row?*

There are two different regions listed for each period in Fig. 3(a). >12h before entering the site in the first line, <12 h in the second line. **We changed "row" to "line" to avoid misinterpretation.**

*Line 287 caption: "absolute number and number fraction"*

**Changed accordingly.**

*Table 2: bottom row, "Local"*

**Corrected.**

*Line 405: Refer to the name of this PAH class- HMW?*

**Changed accordingly.**

*Figure 8: Concerning the acidity aspect- it is interesting that there is no detectable signal for ammonium in these sulfate-containing particles. Although this appears to be the case for several of the PAH classes. Is detection of ammonium less efficient in this system relative to single-step LDI instruments. This would be worth discussing.*

Thank you for this interesting idea and comment! As a rough trend, we find higher ammonium signals in particles with strong signals of (secondary) nitrate (e.g. Figs. 7 and S7) and nearly no ammonium in fresh particles (Fig. 9) and particles from wood combustion (Fig.6.). However, the ammonium signals in PAH-containing particles are generally low and not sufficient for a statistically sound analysis here. We discuss possible effects of the two-step method on the detection of nitrate, sulfate (also valid for ammonium) in the manuscript:

„…Thirdly, the ionization technique itself may contribute to this trend: Particles that are fully hit by the desorption laser produce more intense PAH spectra via REMPI of the plume. However, secondary nitrate or sulfate is also desorbed to a larger extent from these particles, and less amount of this material remains on the particle for detection via LDI in the second laser shot. Nevertheless, in so-far unpublished field experiments with the same setup during winter in Central Europe, we often found PAHs on more than 20% of all particles, also if strong nitrate- and sulfate signals were present, thus the instrumental aspect seems to be of minor importance here."

In several experiments, we could not find a substantial loss of efficiency for such particle components compared to single-step LDI. However, in a direct comparison between single-step LDI at 248 nm and 193 nm wavelength, we see a higher detection efficiency for nitrogen-containing compounds with the 193 nm laser (Passig et al., 2020).

*Citation: https://doi.org/10.5194/acp-2021-722-RC1*

---

## Author Comment (AC2)

Response to Anonymous Reviewer #2
We thank the reviewer for his/her work and the comments. We are feel that addressing the issues raised by the reviewer helped to improve the manuscript. Please see our reply below.

Note:
*Reviewer comments are in italics.*
Author responses are in normal format.
**Changes** that were made to the manuscript are in **bold** face.
* * *
*Review of the manuscript acp-2021-722 with title: "Single-particle characterization of polycyclic aromatic hydrocarbons in background air in Northern Europe" by Passig et al.*

*This manuscript describes the first field application of a recently developed mass spectrometric method (Schade et al., 2019) to analyse single aerosol particles for characteristic components including polycyclic aromatic hydrocarbons (PAH). This is a substantial contribution demonstrating how this new analytical tool goes beyond conventional single particle mass spectrometry to investigate atmospheric aerosol particles on a single particle basis and hence their internal mixing, aging, and potential sources. The combination of mass spectra from laser desorption ionization (LDI) and resonance enhanced multi photon ionisation allows for a more specific assignment especially of combustion related aerosol particle sources. Most of the methods are clearly outlined and the paper is well structured and written. However, some of the results could have been discussed in somewhat more detail. Overall, this manuscript should be accepted for publication after improvements focussing on the specific comments below.*

We thank the referee for carefully reviewing our manuscript, the positive response and the valuable comments.

*Page 1 line 26: The appearance of Calcium in aerosol particles is not generally associated with traffic emissions. Therefore, you should reformulate this to avoid misunderstandings.*

The SPMS-typical aspect of combinations of signatures was added to emphasize that Calcium alone is not a marker here. "…whose inorganic content indicates traffic emissions, such as **combinations of** soot, iron, and calcium…"

*Page 3 line 76: Please give sufficient credit to previous work e.g. by Morrical et al., 1998 who showed one of the first applications of the two-step approach including PAH.*

We thank the reviewer for taking care on credit to previous work, which is an important responsibility for us. However, this sentence focuses on the application of the two-step approaches to ambient air aerosols, which was not the case for Morrical et al. 1998 (laboratory experiments). Instead, the work by Morrical et al. is already referred to in section 3.3.3, p.11.

*Page 3 line 90: Please use add the type and mass resolution of the mass spectrometer.*

Information added: **"The SPMS instrument comprises two reflectron time-of-flight mass analyzers (Stefan Kaesdorf GmbH) and corresponds to the ATOF-MS technique (Prather et al., 1994; Hinz et al., 1994)."** and later in the text: **"The TOF is tuned to**

**achieve a sufficient mass resolution of about R ≈ 800…1000 for both low-mass ions from LDI and high-mass ions from REMPI."**

*Page 3 line 100: Give the size dependent detection efficiencies and discuss their relevance for your results. E.g. to what extend would you miss PAH in smaller particles?*

This information is provided later, in section 2.3, where the sampling is discussed. We added more information there: **"…concentration factor for ambient air particles around 0.5 µm size was estimated to approximately 10:1 and rapidly drops to 1 below 0.5 µm as estimated in a previous study (Passig et al., 2020). Because of the limited detection efficiency for particles below ~150 nm size in SPMS (Su et al., 2004; Zelenyuk et al., 2009) and the size-dependent enrichment factor in this experiment, our study targets on long-range transported particles rather than the ultra-fine size mode."** The added references (Su et al., 2004; Zelenyuk et al., 2009) also emphasize the general limits of optical particle detection in SPMS.

*Page 4 line 122-123: Give the concentration factors for the whole size range of particles measured. Indicate if and how this has impact on the interpretation of your results.*

Thank you! See reply to previous issue. In addition, we inserted a statement on the limitations of the current setup and an outlook for future studies on smaller particles in the conclusions: "Future studies including quantification approaches and simultaneous operation of established on-line technologies such as the Aerodyne AMS (Canagaratna et al., 2007) are planned, **as well as experiments targeting on the PAH distribution in smaller particles, e.g. from local sources and particle formation events, which can hardly be detected with the current setup."**

*Page 5 line 129: Mention that you compare to an optical particle counter.*

Ok, added: **"Aerosol mass concentration was measured with an optical particle counter (Grimm EDM-180 MC) that belongs to the standard instrumentation of the station."**

*Page 5 line 141-144: Explain the manual clustering criteria already in section 2.2 and justify why no automatic procedure was used.*

We desisted from automatic regrouping here and performed the established and safe manual regrouping. We added a statement on the reason: **"Thus, clusters resulting from mass spectral artefacts such as occasional peak broadening or saturation can be evaluated as well as potentially interesting minor clusters."**

*Page 6 Figure 2: Enlarge the mass spectra to the full-page width. Otherwise, they are not readable.*

**Figure enlarged.** Please note that the final production files will contain high-resolution **vector graphic figures in full size.**

*Page 6 line 161: Also smaller particles can contain substantial amounts of secondary material. Specify the sampling and detection bias for different particle sizes and classes in the method section.*

Thank you. See issue for Page 3 line 100 and the reply above for the bias of different particle sizes. Please note, that determining the particle class-dependent detection bias requires extensive laboratory studies (Shen et al. 2018, Shen et al. 2019a), which would exceed the focus of the current study.

*Page 6 line 163-166: Give the fraction of aged sea salt. Compare e.g. to Geng et al., 2010.*

We added statements on the desired information: **"…fresh sea salt (>50% during high local wind speed), aged sea salt (dominant at calm periods)…"**

*Page 7 line 173: Explain why no mineral dust was observed and compare e.g. with Marsden et al., 2018.*

We added a statement on the most plausible reason for the absence of mineral dust as well as a reference to the interesting work of Marsden et al.: **"Mineral dust (Marsden et al., 2018) was not observed, probably because the region is dominated by marine environments and forests in a wide radius."**

*Page 7 line 183: Compare the particle classes with those identified in previous studies at remote locations in Europe and justify your assignment e.g. Lacher et al., 2021, Schmidt et al., 2017, Geng et al., 2010*

With our focus on PAHs, we did not discuss all particle sub-types that can be derived from our LDI data, e.g. the different types of particles associated with sea salt. Instead, we discuss the combustion-related aerosols and their signatures in more detail, as it turned out that they are the important PAH carriers. We added a statement on this in the text and refer to the studies of Schmidt et al. and Lacher et al., which show state-of-the-art analyses of LDI-based particle data in Europe. **"As our study focuses on PAHs and long-range transported combustion aerosols, our LDI-based classification separates into fewer particle types compared to recent studies in Europe (Schmidt et al., 2017; Lacher et al., 2021). Also the limited timeframe of our study and the comparable remote environment with little industrial contribution limit the number of different particle types."**

*Page 7 line 187: Name the instrument optical particle counter.*

Changed accordingly: **"…measured with the optical particle counter of the monitoring station."**

*Page 7 line 198-199: Reformulate this sentence to avoid misunderstanding. Please discuss if this could also be influence e.g. by a lower detection efficiency for sulphate rich particles?*

Particles with dominant sulfate signals (sulfur-rich) were frequently detected in SPMS studies in marine environments. Important sources are ship emissions and marine life. We added corresponding SPMS studies showing and discussing this particle type and reformulated the sentence accordingly: **"Despite the marine origin, only a few particles with dominant sulfate signals in anion spectra were detected (Dall'Osto et al., 2016a; Arndt et al., 2017; Wang et al., 2019). Possible reasons comprise the sulfur limits for ship fuels and low marine biogenic activity in autumn."** There is no evidence to assume a lower detection efficiency for our method compared to other SPMS studies detecting these particle types.

*Page 7 line 201: Please clarify what you mean with sulphur containing and sulphur rich particles.*

The term "sulfur-rich" is limited to particles with dominant sulfate peaks in anion mass spectra, while "sulfur-containing" refers to particles with smaller sulfate signals. This follows the termination of several SPMS papers which are now referred to at the respective sentence (see also previous issue) **"dominant sulfate signals in anion spectra were detected (Dall'Osto et al., 2016a; Arndt et al., 2017; Wang et al., 2019)."**

*Page 8 line 214-215: How did you identify night-time new particle formation?*

We attribute the occasional strong increase of particles during nighttime to this effect, e.g. at the $17^{th}$ Oct. As we agree with the reviewer that there is no sufficient evidence to estimate the role of this effect, we reformulate the sentence accordingly: **"…numbers as well as transient features, e.g. from possible night-time particle formation events at light winds from land and along the coastline."**

*Page 8 line 216: Can you really give relative contributions of different particles classes? Do you account for different detection efficiencies for different particle classes? Please discuss this addressing e.g. Shen et al., 2019a.*

We performed a conventional SPMS approach, where particle numbers are not corrected for inlet, (optical) detection and ionization efficiencies and state this in our manuscript. Such corrections require detailed laboratory data on the instrumental response to all types and sizes of particles relevant in the respective study, considering effects of actual atmospheric conditions (humidity), secondary material and matrix effects of the respective ionization method. However, we added two further references to the interesting studies of Shen et al. and the general possibility to apply such corrections: **"…while Fig. 3(d) shows the same data normalized to total particle counts without corrections for particle detection efficiencies as performed by (Shen et al. 2018) and (Shen et al., 2019a)."**

*Page 8 line 222: Which evidence do you have for this?*

Please see issue on Page 8 line 214-215 and reply above.

*Page 8 line 225: There is no data shown for October $14^{th}$.*

Thank you. Typo corrected: **"…first period of marine air and strong wind ($12^{th}$ -$13^{th}$ of October),…"**

*Page 9 line 233: …peak area….*

Thank you. **Corrected.**

*Page 11 line Figure 4: Please enlarge the mass spectra as to make them readable.*

**Corrected (enlarged).** Please note that we prepared **high-resolution files for production**.

*Page 12 line 315: Please reformulate this sentence, as iron is not increasing during transport.*

Reformulated: "The **larger** fraction of Fe-containing particles **during periods of air mass transport** from…"

*Page 12 line 322-323: Please reformulate as you did average the mass spectra but you did not mix them.*

Changed accordingly: **"…mixed → averaged in…"**

*Page 12 line 324: 53% of the PAH containing particles were not classified….*

To improve clarity, **we reformulated, how particle were counted** and provided the percentages: "A substantial fraction of PAH-containing particles is not classified concerning their inorganic composition **because they either produced now LDI mass spectra (21%)** or belong to clusters with small particle numbers and high orders beyond 300 **(42%),**…"

*Page 12 line 325-326: Please reformulate. E.g. Their mean PAH spectrum originates from different particle types… .*

Changed: **"…mixed → results from…"**

*Page 13 line 345: Explain the different number of PAH containing particles compared to Figure 4a.*

**We reformulated the counting information** for PAH-containing particles with additional LDI information, see issue for Page 12 line 324 above. Thus, this inconsistency is resolved. The REMPI-based clustering considers the full ensemble of particles with PAH signatures, also particles without LDI signals.

*Page 14 Table 2: Correct "Local green" in row 'alkylated LMW'.*

Thank you for the careful check! **Corrected accordingly.**

*Page 15 line 379: Explain the meaning of 'Ox'.*

Information added: "Ion mass channels: EC: 24–84/36, 48; OC: 42/27, 29, 55, 63; Ox: 45, 59, 71/43 **(oxygen-containing fragments);** K: -/39; Nit: 46, 62/-; Sul: 97, 99/-; Fe: -/54, 56; Ca: -/40; Phos: 63, 79/- for negative/positive m/z, respectively).

*Page 15 Figure 5c: The grey dots are not good visible.*

**Changed to a darker grey improving contrast.**

*Page 15 Figure 5d: Choose fillings or colours that allow better to distinguish between the particle classes.*

**We changed the colors in all Figs. 5-9 and S3-S7 to improve contrast.**

*Page 15 line 388: Explain the criteria for manual classification of the subgroups and compare them with the classification in section 3.1.*

Thank you! We added a statement on the necessary data reduction and focusing on the most important features of the LDI data here: "…the results were manually merged to **six subgroups, each representing the key characteristics of the particle composition from LDI .**" Please note, that the current section is crucial for the reader to follow the analysis of this deep and complex dataset, with the third clustering procedure of this study described here. Thus, we aim on a concise writing style to allow the reader to follow the complete story and to get a first insight into the key features and the method's potential rather than a full in-depth analysis of the dataset.

*Page 15 line 399-400: …among most particle subgroups….*

**Corrected.**

*Page 17 line 435-448: Please give an estimate of the transport times from potential source regions and typical PAH degradation time scales for typical atmospheric conditions. Demonstrate that your interpretation is reasonable.*

This is an important aspect and points towards one of the potential key applications of the developed technology. Degradation and long-range transport of PAHs cannot be well quantified by conventional laboratory studies and are a hot topic in atmospheric chemistry. We added a short discussion on degradation and some references. The transport times from source regions in our study can be estimated from Figs. 3a and S2. **"PAH degradation in airborne particles is strongly affected by matrix effects and varying atmospheric conditions. Laboratory experiments can hardly represent the matrix of organic and inorganic secondary material and possible shielding effects (Zelenyuk et al., 2012; Abramson et al., 2013; Shrivastava et al., 2017; Alpert et al., 2021). PAH lifetimes for homogeneous reactions with oxidized N species are reported between 1 hour and 2 days, while lifetimes towards heterogeneous photochemical degradation are estimated between few hours and >10 days, both for typical continental background conditions in mid latitudes, see (Keyte et al., 2013) end references therein. Long-range transport of PAHs is well documented, while concentrations measured at remote sites are typically more than one order of magnitude lower than within the respective source regions (Jaward et al., 2004). Evaluation of single-particle PAH signatures may therefore provide a rough estimate on degradation, if the source profile is known or indicated by the inorganic particle composition in a sophisticated future measurement scenario."**

*Page 18 line 452: Do you mean: 'REMPI spectra of several PAH classes are…'.*

Yes, thank you. **Changed accordingly.**

*Page 19 line 505: Please discuss the LDI spectra.*

Discussion on LDI mass spectra added: **"Many of these particles show dominant K+ peaks that are characteristic for biomass- and wood burning emissions (Silva et al., 1999). Also nitrate and especially sulfate signals are small, as expected for relatively fresh particles."**

*Page 19 line 515: Please use approximately instead of approx. in the text.*

**Corrected.**

*Page 19 line 516: …was previously observed in the analysis of….*

**Corrected accordingly.**

*Page 20 line 538: This statement is only correct if you could quantify the individual particle classes.*

We understand the issue and thank the reviewer for initiating a fruitful discussion. However, we did not claim to quantify the PAHs nor to provide a detailed quantification of the PAH load in each particle class. Here, we only state to have identified the major PAH carriers. Please note that the relative fraction of PAH-containing particles differs substantially between the particle classes, from 0% for sea salt to 12% for fresh soot (See Fig. 4). The Fe-Nit-OC class alone accounts for about one half of the (LDI-characterized) PAH containing particles, while the even larger class of Fe-Nit-Soot particles shows only 65 of such particles. We are convinced that corrections of the inlet, detection and ionization efficiency (e.g. Shen et al. 2019a) would not change this picture completely.

*Page 20 line 548: Comparison with additional measurements, e.g. those you have already done, would probably help to do a systematic analysis for a more reliable source apportionment. However, also a comparison with dedicated transport model calculations could help to substantiate you interpretations.*

We fully agree with the reviewer. Systematic studies on source profiles are on the way. However, previous studies on single-particle PAH signatures (without determining the inorganic composition) revealed already some typical source-specific PAH profiles and are referred to in the manuscript (Morrical et al., 1998; Bente et al., 2008, 2009; Li et al., 2019). Transport modelling is a powerful tool that will be integrated in future studies. We added a comment on that. "Future studies **including transport modelling for reliable determination of source regions**, quantification approaches…"

*Page 20 line 549: If you do this kind of measurements for the first time it can be expected that you would have taken care for suitable reference measurements either yourself or by inviting suitable other groups.*

Some reference measurements are published in the article introducing this method (Schade et. al. 2019). There are laboratory studies on many of the PAH-signatures discussed in this manuscript (Morrical et al. 1998, Bente et al. 2008, 2009, Czech et al. 2017, Liu et al. 2019). All LDI signatures were conservatively interpreted and literature is provided. New is the application of this method integrating LDI- and REMPI-based information on a complex ambient air aerosol as well as the analysis approach, both without precedent. However, studies on specific sources and ageing mechanisms are planned. We added an outlining statement on planned experiments that also emphasizes important limitations of the current study: **"…the as well as experiments targeting on the PAH distribution in smaller particles, e.g. from local sources and particle formation events, which can hardly be detected with the current setup."**

*Page 21 line 555-558: If you have already a larger database for a more systematic and statistically relevant analysis, wouldn't it be possible to make use of it to achieve a better interpretation of the data collected during the measurements described in this manuscript?*

The newer database is not directly comparable, as it is a winter measurement in a different region. Although more particles contain PAHs, the ensemble is more homogeneous and many

of the interesting atmospheric ageing effects described here occurred rarely in the winter measurements. Therefore it will be published in a new paper.

*Page 21 line 560: Consider giving all relevant data for your measurement campaign including the mass spectra of specific particle classes to an open data repository instead of adding 69 pages to the supplement (e.g. https://www.pangaea.de/).*

**We added all cluster spectra and time series in the open data repository at Zenodo and shortened the supplement accordingly** https://doi.org/10.5281/zenodo.5794078.

**References beyond the ones in the manuscript:**

Geng et al., Single-Particle Characterization of Summertime Arctic Aerosols Collected at Ny-Ålesund, Svalbard, Environ. Sci. Technol., 44, 7, 2348–2353, 2010. https://doi.org/10.1021/es903268j .